# A spatially resolved stochastic model reveals the role of supercoiling in transcription regulation

**Yuncong Geng**[1¤a]*, **Christopher Herrick Bohrer**[2,3¤b], **Nicolás Yehya**[3], **Hunter Hendrix**[3], **Lior Shachaf**[2], **Jian Liu**[4], **Jie Xiao**[3], **Elijah Roberts**[2¤c]

**1** Department of Biomedical Engineering, Johns Hopkins School of Medicine, Baltimore, Maryland, United States of America, **2** Department of Biophysics, Johns Hopkins University, Baltimore, Maryland, United States of America, **3** Department of Biophysics and Biophysical Chemistry, Johns Hopkins School of Medicine, Baltimore, Maryland, United States of America, **4** Center for Cell Dynamics, Department of Cell Biology, Johns Hopkins University School of Medicine, Baltimore, Maryland, United States of America

¤a Current address: Center for Biophysics and Quantitative Biology, University of Illinois at Urbana-Champaign, Urbana, Illinois, United States of America
¤b Current address: Laboratory of Receptor Biology and Gene Expression, Center for Cancer Research, National Cancer Institute, National Institutes of Health, Bethesda, Maryland, United States of America
¤c Current address: 10x Genomics Inc., Pleasanton, California, United States of America
* yuncong5@illinois.edu

**Data Availability Statement:** All input data and code is available at https://github.com/gengyuncong/Supercoiling.

## Abstract

In *Escherichia coli*, translocation of RNA polymerase (RNAP) during transcription introduces supercoiling to DNA, which influences the initiation and elongation behaviors of RNAP. To quantify the role of supercoiling in transcription regulation, we developed a spatially resolved supercoiling model of transcription. The integrated model describes how RNAP activity feeds back with the local DNA supercoiling and how this mechanochemical feedback controls transcription, subject to topoisomerase activities and stochastic topological domain formation. This model establishes that transcription-induced supercoiling mediates the cooperation of co-transcribing RNAP molecules in highly expressed genes, and this cooperation is achieved under moderate supercoiling diffusion and high topoisomerase unbinding rates. It predicts that a topological domain could serve as a transcription regulator, generating substantial transcriptional noise. It also shows the relative orientation of two closely arranged genes plays an important role in regulating their transcription. The model provides a quantitative platform for investigating how genome organization impacts transcription.

## Author summary

DNA mechanics and transcription dynamics are intimately coupled. During transcription, the translocation of RNA polymerase overwinds the DNA ahead and underwinds the DNA behind, rendering the DNA supercoiled. The supercoiled DNA could, in return, influences the behavior of the RNA polymerase, and consequently the amount of mRNA product it makes. Furthermore, supercoils could propagate on the DNA over thousands of base pairs, impacting RNA polymerase molecules at faraway sites. These complicated

**Funding:** J.X. and E.R. received funding from National Science Foundation (https://www.nsf.gov/) under grant MCB1817551. J.L. received funding from National Science Foundation (https://www.nsf.gov/) under grant MCB2105837. C.H.B. and N.Y. received funding from National Institutes of Health (https://www.nih.gov) under grant 5T32GM007231. The funders had no role in study design, data collection and analysis, decision to publish, or preparation of the manuscript.

**Competing interests:** The authors have declared that no competing interests exist.

interplays between supercoiling and RNA polymerase makes supercoiling an important transcription regulator. To quantitatively investigate the role of supercoiling in transcription, we build a spatially resolved model that links transcription with the generation, propagation, and dissipation of supercoiling. Our model reveals that supercoiling mediates transcription at multiple length scales. At a single-gene scale, we show that supercoiling gives rise to the collective motion of co-transcribing RNA polymerase molecules, supporting recent experimental observations. Additionally, large variations in mRNA production of a gene can arise from the constraints of supercoiling diffusion in a topological domain. At a multi-gene scale, we show that supercoiling dynamics allow two adjacent genes influence each other's transcription kinetics, thus serving as a transcription regulator.

## Introduction

Recent studies show that in *E. coli* the chromosomal DNA is under torsional stress. A relaxed DNA (with every 10.5 bp per turn [1]) becomes torsionally stressed if it is underwound (negatively supercoiled) or overwound (positively supercoiled). In cells, the chromosomal DNA is held in a homeostatic, negatively supercoiled state globally by the coordination of topoisomerases [2,3]. However, DNA-related processes such as replication and transcription could drive the local supercoiling state away from its equilibrium. For example, the twin-domain model of transcription depicts that during elongation, RNA polymerase (RNAP) overwinds the DNA downstream and underwinds the DNA upstream to generate positive and negative supercoiling respectively, leading to the build-up of DNA torsional stresses in the vicinity of RNAP [4,5]. During this process, RNAP"encodes" torsional information on the DNA. If the torsional stress is not relieved in a timely manner, it could propagate through the DNA and influence the behavior of other DNA-bound RNAP molecules from afar [6]. Conversely, RNAP"retrieves" the torsional information by adapting its initiation and elongation behaviors in response to local supercoiling. Many promoters display supercoiling sensitivity [7,8,9], and high torsional stress could slow down [8] or even stall [10,11] elongating RNAP molecules.

The unique property of supercoiling to store and transmit RNAP-accessible information makes it a medium for RNAP molecules on the same transcription unit or even different operons kilobases away to communicate with each other [12,13]. Therefore, supercoiling has been proposed to serve as a transcription regulator across multiple distance scales from a few to thousands of base pairs, giving rise to many emergent properties in transcription.

At the single-gene scale, supercoiling is proposed to coordinate the collective motion of RNAP molecules during elongation, enabling RNAP to move at different speeds with different initiation rates. The promoter strength, in this case, emerges as a regulator for the elongation speed. Kim *et al.* [13] found that strong promoters facilitate the processivity of RNAP when the promoter is on and inhibits the processivity of RNAP when the promoter is off. The authors hypothesized that during transcription, opposite supercoils generated in the region between two adjacent RNAP molecules can cancel out with each other quickly, reducing the torsional stress in the region and hence making the translocation of both RNAP molecules faster.

At the multi-gene scale, supercoiling could mediate the interaction between multiple closely arranged genes. Genome-wide evidence of correlation of transcriptional activity between neighboring genes has been detected [14, 15], and the transcription of gene cassettes inserted in the *E. coli* genome was found to depend strongly on the transcription activity of adjacent genes at the insertion site [16,17]. It is suggested that supercoiling plays a role in producing the

correlation—transcribing RNAP molecules of adjacent genes keep rewriting the intergenic supercoiling profile and preventing their dissipation, hence enabling neighboring genes to affect the transcription of each other by sharing the same torsional stress state [14,15,16].

Supercoiling could also regulate the transcription of genes belonging to the same topological domains. In *E. coli* cells, there are about 400 topological domains of an average size of ~10 kb [18, 19], which are likely formed by the dynamic looping and unlooping of the intervening DNA due to the binding and unbinding of nucleoid-associated proteins. Those proteins, by the nature of their binding to the DNA, restrict the diffusion of supercoiling and hence constrain torsional stress within topological domains. Supporting this possibility, theoretical work has shown that the opening and closing of a topological domain could lead to transcriptional bursting [20] and correlate gene transcription in the same domain [21].

Although supercoiling is proposed to mediate the communication between multiple RNAP molecules and multiple genes, it is not clear how supercoiling dynamics lead to changes in transcription activities quantitatively and how these changes are dependent on the mechanical properties of DNA and the transcription kinetics of RNAP. Several models [20,21,22,23,24,25,26,27,28] have been established to explore the role of supercoiling in transcription regulation. However, none of these models fully captured the complex interactions between transcription and supercoiling. Instead, they mainly focused on a subset of key aspects, such as the contribution of supercoiling to transcription initiation [21,23,24,26] or elongation [20,22,25,27] only, and supercoiling diffusion was not explicitly modeled [28]. Here we present a spatially resolved, chemical master-equation based model of *E. coli* transcription that explicitly describes all stages of transcription (initiation, elongation, and termination, **Fig 1A–1C**), and their interplay with DNA supercoiling and other cellular processes that regulate DNA torsional stress (topoisomerase activity, supercoiling diffusion, and the dynamical formation and dissociation of topological domains, **Fig 1D–1G**). This comprehensive model integrates the most recent experimental observations of the relationships between transcription and supercoiling [8,10,11,29,30,31].

Using this model, we established that inter-RNAP supercoiling dynamics could quantitatively explain the cooperation of co-transcribing RNAP molecules evidenced in experiments, and the degree of cooperation is sensitive to supercoiling diffusion rate and topoisomerase activities. We also predicted that by confining supercoiling diffusion, a topological domain could give rise to bursty transcription of genes within the domain. Furthermore, transcription-induced supercoiling leads to complex communications between two closely arranged genes. Our findings provide insights into how supercoiling regulates gene expression with different promoter architectures and gene arrangements, paving the way for quantitative studies of how genome organization impacts gene expression.

## Methods

### General design of the model

To interrogate the role of supercoiling in transcription, the model describes how supercoiling feeds back with RNAP's transcription activity and how this feedback evolves as a reaction-diffusion process along the DNA length to coordinate the transcription of multiple RNAP molecules and different genes within the DNA topological domain (**Fig 1**). Briefly, there are three key model ingredients. First, while topoisomerases relax DNA supercoiling and keep it in balance, RNAP-mediated transcription introduces DNA torsional stress and hence change the extent of DNA supercoiling. Second, alteration in DNA supercoiling, in turn, impacts the initiation and elongation rates of RNAP. Therefore, interactions between transcription and DNA supercoiling form a mechanochemical feedback. Last and most importantly, DNA torsional

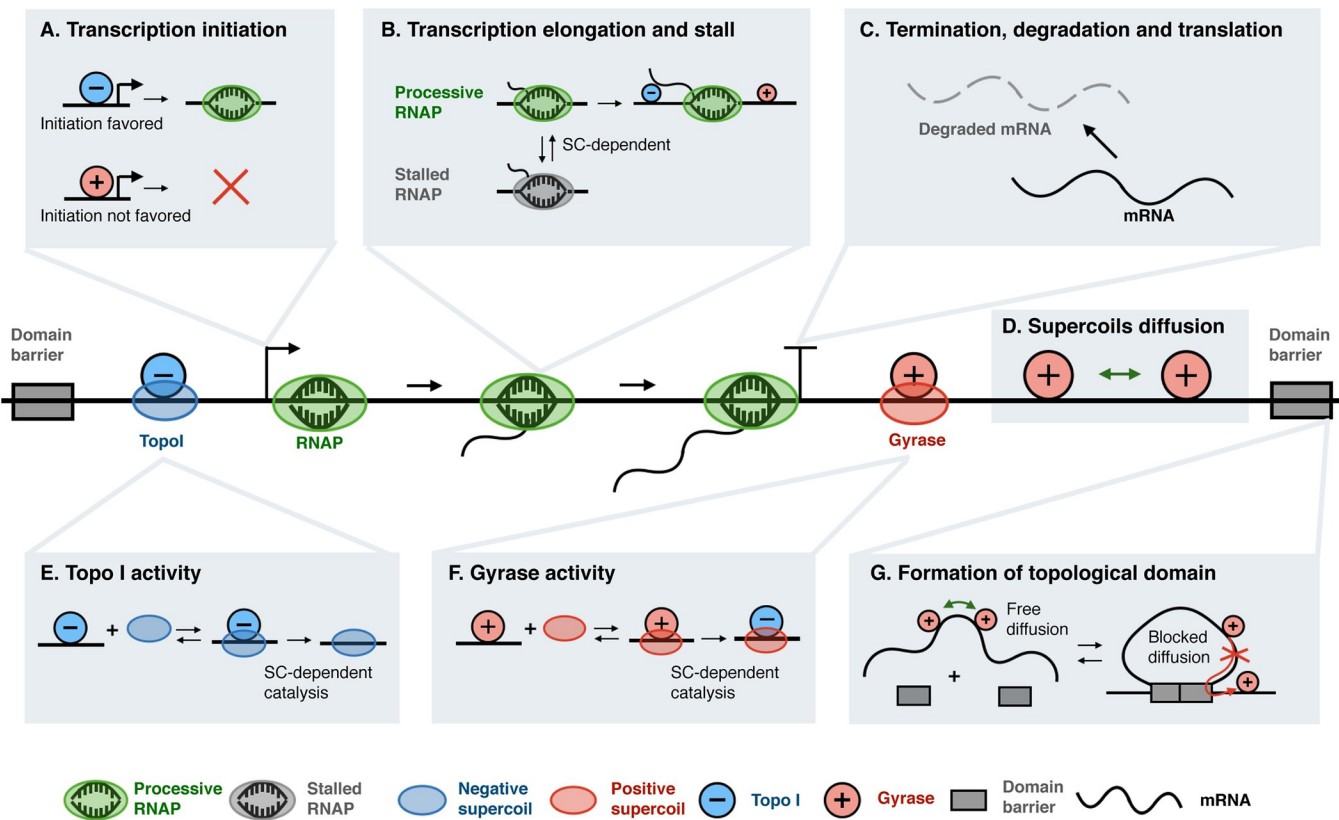

**Fig 1. Model schematics of supercoiling-dependent transcription.** (A) During transcription initiation, a negatively supercoiled (blue circle) promoter is favored over a positively supercoiled (red circle) promoter for RNAP (green oval) binding and open complex formation [7]. (B) During transcription elongation, the translocation of RNAP induces positive supercoiling in front and negative supercoiling behind [4,5], which consequently influences whether the RNAP remains processive (green) in elongation or becomes stalled (grey) [10,11]. (C) Transcription terminates when RNAP is released from the mRNA. The transcribed mRNA is subsequently degraded in the model. (D) Supercoils can diffuse on DNA within a topological domain flanked by two domain barriers and interact with RNAP and topoisomerases. (E) Topo I (blue oval) removes one negative supercoil at a time [32]. (F) Gyrase (red oval) converts one positive supercoil to one negative supercoil at a time [31]. (G) Formation and dissolution of a topological domain upon the binding and unbinding of domain anchoring proteins (domain barriers, grey block).

stress diffuses along the length of the DNA and is limited by the topological barrier. By integrating the three ingredients into a coherent model of transcription, we explore below how this mechanochemical feedback controls the spatial-temporal coordination between the transcription of multiple RNAP molecules and different genes.

To quantitatively capture the model essence and the stochastic nature of the process, we exploited the mathematical formation of spatially resolved chemical master equation to delineate the reaction-diffusion process of the mechanochemical feedback. Specifically, we treated the DNA as a linear 1-D lattice by dividing the DNA into a series of 60-bp segments; the choice of 60-bp is to a compromise to accommodate the length scale of both RNAP footprint (32 bp) [33] and gyrase binding site (137 bp) [34] and to minimize computational costs. Supercoils, RNAP or topoisomerase molecules at different positions of the lattice were considered as different species. This strategy enabled us to describe the spatial distribution of RNAP, topoisomerases, and supercoils at different positions along the DNA over time. We performed simulations with exact sampling (Gillespie algorithm) in Lattice Microbes [35], with counts of species recorded every 1 s. Due to the limited number of species allowed in Lattice Microbes, the maximum size of DNA that could be simulated is ~ 20 kb. The detailed rationale of this strategy can be found in **S1 File**.

**Table 1. Timescale for key processes in supercoiling dissipation and generation.**

| Categories | Process | Parameter | Timescale |
|---|---|---|---|
| Supercoiling diffusion | | Undetermined; Various diffusion coefficients (D) from $0.002 \sim 2\ \mu m^2 \cdot s^{-1}$ are tested in this study | $\sim 10^{-4}\ to\ 10^{-1} s$ (calculated by $\frac{(60bp)^2}{2D}$) |
| Processes that contribute to supercoiling dissipation | Gyrase unbinding | $k_{unbind} \sim 0.5\ s^{-1}$, directly measured by [30] | $\sim 2$ s (calculated by $\frac{1}{k_{unbind}}$) |
| | Gyrase binding | $k_{bind} \sim 0.002 s^{-1} \cdot (60\ bp)^{-1}$, estimated from [30] | $\sim 500$ s (calculated by $\frac{1}{60bp \cdot k_{bind}}$) |
| | Topo I unbinding | Undetermined; Various $k_{unbind}$ from $0.001 \sim 100\ s^{-1}$ are tested in this study | $\sim 0.01$ to 1000 s |
| | Topo I binding | Undetermined; Assumed to be the same as Gyrase | $\sim 500$ s |
| | RNAP rotation | Undetermined; $k_{rot}$ set as $0.2\ s^{-1}$ in this study | 5 s |
| | Domain unlooping | Undetermined; Looping rate set as $0.017\ s^{-1}$ in this study | 60 s |
| Processes that contribute to supercoiling generation | Transcription elongation | $k_{elong} \sim 60\ bp/s$ [36] | $\sim 1$ s (calculated by $\frac{60bp}{k_{elong}}$) |
| | Domain looping | Undetermined; Unlooping rate is set as $0.003\ s^{-1}$ in this study | 300 s |

To describe the dynamics in the coupling of transcription and supercoiling, we focused on two questions: (1) how DNA supercoils are generated and dissipated by transcription and other processes, and (2) how transcription responds to DNA supercoiling. To address the first question, we incorporated into our model three processes that are known to contribute to the dynamics of DNA supercoiling: (a) diffusion of supercoils, (b) activities of topoisomerases, and (c) transcription. Topoisomerases regulate supercoiling globally, transcription perturbs supercoiling locally, and through diffusion, those changes are transmitted along the DNA. The timescale for key processes in supercoiling dissipation and generation are shown in **Table 1**. To address the second question, our model considered supercoiling-sensitive transcription initiation and elongation. Detailed reaction schemes are described below. All model equations and parameters can be found in **S1–S3 Tables**.

## Modeling diffusion of supercoiling

DNA torsional stress can be stored in the form of either twist or writhe [37]. Twist is the rotation of DNA around its rise axis. Writhe is the cross-over of DNA double-helix with each other. When a sufficiently high torque is applied to DNA, instead of twisting, DNA will wrap around itself to form writhes, since the energetic cost of bending becomes lower than that of twisting [38, 39] (**S1 Fig**). Further increases in the torque will cause the writhes to pile up and form plectonemes [38,39]. Twists and writhes diffuse differently along DNA. Twists diffuse rapidly along the DNA, with an estimated diffusion constant of $50 \sim 180\ \mu m^2 \cdot s^{-1}$ [40,41]. Writhes diffuse slowly, with a very small diffusion coefficient (D) of $0.01 \sim 0.2\ \mu m^2 \cdot s^{-1}$, according to *in vitro* experiments performed by Loenhout *et al.* [29]. In this study, a wide range of D (from $0.002 \sim 2\ \mu m^2 \cdot s^{-1}$) were tested.

In our model, we used "supercoiling density" (σ) to quantify the level of torsional stress, and we used the universal term "turns" to represent both twists and writhes. Supercoiling density is defined as $\sigma = \frac{Lk - Lk_0}{Lk_0}$, where Lk is the actual number of turns in the DNA segment, and $Lk_0$ is the number of turns in the DNA when it is fully relaxed (one turn per 10.5 bp). A relaxed, 60 base pair DNA segment has a $Lk_0$ at 60 bp/(10.5 bp/turn) = 5.7 turns/segment. In the *E. coli* chromosome, the average supercoiling density σ is at -0.05 $\sim$ -0.07 [3]. We used Turn(k) to

track the total number of turns on the k-th DNA segment. Since the quantities in chemical master equations need to be integers, we define the new $Lk_0$ as 57. To initialize a negatively supercoiled DNA with $\sigma = -0.05$, we set Turn(k) = 54 all k at t = 0.

We used DNA($k$) to denote the availability of the $k$-th DNA segment. DNA($k$) = 1 indicates that the DNA is unoccupied and available for the binding of RNAP, topoisomerase or domain-anchoring proteins, and DNA($k$) = 0 indicates that the DNA segment is occupied by a DNA binding-protein and hence unavailable. To reduce the computational load, we modeled the diffusion of turns as a biased random walk, which means that we only allowed turns to move down the gradients:

If Turn($k$)>Turn($k-1$) and DNA($k-1$) = 1 (unoccupied), the turns on the $k$-th segment diffuse to ($k-1$)-th segment:

$$\text{Turn}(k) \rightarrow \text{Turn}(k-1)$$

Similarly, if Turn($k$)>Turn($k+1$) and DNA($k+1$) = 1:

$$\text{Turn}(k) \rightarrow \text{Turn}(k+1)$$

The rate at which turns diffuse along the DNA segments is determined by the propensity of the above two reactions and the rate constant for turn displacement, $k_{drift}$, where $k_{drift} = \frac{2D}{(60\ bp)^2}$.

## Parameterizing the binding kinetics and catalytic activities of topoisomerases

We decomposed topoisomerase activities into three steps: binding to DNA (with the rate constant of $k_{bind}$), catalysis (with the rate constant of $k_{cat}$), and dissociation (with the rate constant of $k_{unbind}$). We assumed that topoisomerase binds indiscriminately throughout DNA, and that the catalysis rate depends on the local supercoiling density. Instead of modeling the binding of individual topoisomerase molecules explicitly, we assumed a pseudo-first-order topoisomerase binding reaction for each DNA segment. Using Gyrase as an example, "Gyrase_bind(k)" and "Gyrase_unbind(k)" denote the binding and unbinding state of the $k$th DNA segment, respectively. We initialized the system with "Gyrase_unbind(k) = 1" at all the unoccupied DNA segments, and used the following equation for Gyrase binding, with rate constant $k_{bind}$:

$$\text{DNA(k)} + \text{Gyrase\_unbind(k)} \rightarrow \text{Gyrase\_bind(k)}$$

The unbinding is described by the following reaction with rate constant $k_{unbind}$:

$$\text{Gyrase\_bind(k)} \rightarrow \text{DNA(k)} + \text{Gyrase\_unbind(k)}$$

Below we deduced the values of the three rate constants from the existing data. Stracy *et al.* [30] recently characterized the binding kinetics of Gyrase in *E. coli* using single-particle tracking. There are about 300 Gyrase molecules stably bound to the DNA at any time, and the average dwell time is about 2 s. Since the *E. coli* genome is about 4 Mb, taking a deterministic approximation, we have $k_{unbind} = 1/(2\ s) = 0.5\ s^{-1}$, and the pseudo-first-order binding rate $k_{bind}$ = (300/(4 Mb/(60 bp/segment)))/(2 s) = 0.00225 $s^{-1} \cdot segment^{-1}$. Because of the deterministic approximation, we further corrected $k_{bind}$ to 0.0018 $s^{-1} \cdot segment^{-1}$ to match published experimental results [30]. For Topo I's binding kinetics, no direct experimental measurements are available. However, since topA (subunit for Topo I) and gyrA (subunit for Gyrase) have comparable copy numbers in the cell [42], we assume that the binding rate of Topo I is the same as that of Gyrase, and we treat the unbinding rate as a free parameter. The response of catalysis activity to supercoiling density $k_{cat}(\sigma)$ for both Topo I and Gyrase was parameterized from the DNA relaxation assays (**S1 File**) [8,31].

## Modeling the effects of transcription on supercoiling

In *E. coli*, a transcribing RNAP molecule usually forms a macromolecular complex with a nascent mRNA molecule, translating ribosomes, and newly translated peptides. As such, a transcribing RNAP molecule is, in general, considered as a topological barrier that blocks the diffusion of supercoiling. A transcribing RNAP molecule can travel linearly along the DNA to generate torsional stress, and/or counterrotate to release torsional stress. Our model considers both situations as described below.

Assuming that an RNAP molecule does not rotate at all, the linking number upstream and downstream of it would be conserved. As a result, when an RNAP molecule travels along the DNA, it will push all the turns downstream forward instantaneously and introduce torsional stress (**S2 Fig**). Since RNAP displacement is powered by the chemical reactions of NTP's incorporation, and the concentration of NTP is held constant in most cases, we expected a constant translocation rate (60 $bp·s^{-1}$) [36]. To avoid collisions between multiple RNAP molecules and the turns of each DNA segment pushed by RNAP, we only allowed displacement of RNAP molecules to occur when the two downstream segments (namely, DNA(k+1) and DNA(k+2)) are not occupied by other RNAP molecules.

In addition to directly displacing turns on adjacent DNA segments, RNAP could also counterrotate itself to release torsional stress [8]. In the *in-vitro* study by Chong *et al.* [8], it was observed that the transcription initiation rate of a circular DNA remained constant in the absence of topoisomerase even when both ends were fixed to a surface to avoid free rotations. In this case, as an RNAP molecule approaches the end of a gene that is fixed to the surface by biotin tags (serving as topological barriers), an extreme level of positive supercoiling would build to the extent that stalls transcription elongation and subsequently initiation if RNAP does not counterrotate to release the stress. Premature dissociation of RNAP could also be one way to release the torsional stress, but it is unlikely because it is well known that the transcription elongation complex is extremely stable and unlikely to dissociate spontaneously without the help from active release factors *in vivo* [43]. Here we modeled the counterrotation of RNAP as the diffusion of supercoils over RNAP (**S2 Fig**). For any RNAP(*k*) = 1, if Turn(*k-1*)> Turn(*k-1*) and DNA(*k-1*) = 1, the turns on the (*k+1*)-th segment diffuse to (*k-1*)-th segment:

$$Turn(k + 1) \rightarrow Turn(k-1)$$

Similarly, if Turn(*k-1*)>Turn(*k+1*) and DNA(*k+1*) = 1:

$$Turn(k-1) \rightarrow Turn(k + 1)$$

The rate constant is defined by $k_{rot}$, which we kept as a free parameter. We also assumed that RNAP's displacement and rotation are uncoupled.

## Modeling supercoiling-sensitive transcription initiation

Open complex formation is a crucial step in transcription initiation, and promoter-melting is favored by negatively supercoiled DNA [7]. Recently, using single-molecule fluorescence *in-situ* hybridization (smFISH), Chong *et al.* [8] confirmed that both the T7 RNAP and the *E. coli* RNAP have supercoiling-sensitive initiation rates. These experiments motivated theoretical studies to quantify the influence of supercoiling on transcription initiation. By adopting a statistical mechanics model, Bohrer *et al.* [21] approximated the initiation rate $k_i(\sigma)$ by a linear function of the supercoiling density $\sigma$ at the promoter. Although this model quantitatively recapitulated the dynamics of the initiation rate observed by Chong *et al.* [8], it does not constrain the initiation rate. We leveraged a truncated version of this formulation here to describe

the supercoiling-sensitive initiation:

$$k_i(\sigma) = \begin{cases} k_{min}, \sigma \geq 0 \\ k_{min} + (k_{max} - k_{min})\dfrac{\sigma}{\sigma_*}, \sigma_* < \sigma \leq 0 \\ k_{max}, \sigma \leq \sigma_* \end{cases}$$

Where $\sigma^*$ is a critical level of negative supercoiling density at which the initiation rate reaches its maximum. The initiation rate reaches its minimum when DNA is fully relaxed or further positively supercoiled. In the following simulations, $\sigma^*$ is held at -0.06, and the minimum initiation rate $k_{min}$ is held as 0.001 $s^{-1}$, corresponding to 1~2 initiation events per generation (20 min). The maximum initiation rate $k_{max}$ is allowed to vary up to 0.2 s$^{-1}$ for different promoters.

## Modeling supercoiling-sensitive transcription elongation

During transcription, the change in downstream and upstream DNA supercoiling generated by RNAP translocation will introduce torque around the RNAP [10]. Our model allows RNAP to switch between two modes (stalled and processive) [44], where the translocation velocity (or elongation speed) is 60 bp/s in the processive mode and 0 bp/s in the stall mode. When the torque is beyond a threshold, RNAP enters the stall mode instantaneously. After the torque drops below this threshold, RNAP returns to the processive mode. Ma *et al.* [10, 11] has shown that *E. coli* RNAPs can endure torques up to about 10.5 $pN \cdot nm$. We used this value as the stall torque threshold. To calculate the torque for each RNAP, we first calculated the supercoiling density of its immediate upstream and downstream DNA, and then converted the supercoiling density at each side to torque by the formulation proposed by Marko [45]:

$$\tau = \begin{cases} \dfrac{c_s}{\omega_0} \cdot \sigma, |\sigma| \leq |\sigma_s| \\ [2pg/(1-p/c_s)]^{\frac{1}{2}}/\omega_0, |\sigma_s| < |\sigma| < |\sigma_p| \\ \dfrac{p}{\omega_0} \cdot \sigma, |\sigma| \geq |\sigma_p| \end{cases}$$

For each RNAP molecule, the overall torque it faces is calculated from the downstream torque minus the upstream torque. The detailed parameters can be found in **S1 File**.

Besides torque, we also assumed that an elongating RNAP molecule could stall under other circumstances caused by extreme supercoiling densities. For example, when the upstream DNA is hyper-negatively supercoiled, R-loop might form between the template strand and the nascent mRNA [46], and hence arrest RNAP [47]. Additionally, when downstream DNA is hyper-positively supercoiled, the free energy required to melt DNA could be extremely high for RNAP to proceed [8]. To account for these effects, we set $\sigma = \pm 0.6$ as the supercoiling density threshold that leads to stalled RNAP.

## Modeling looping and unlooping

We modeled the transition between looped and unlooped states, with transition rates $k_{loop}$ (from unlooped to looped state) and $k_{unloop}$ (from looped to unlooped state). We designated two DNA segments as the binding sites for the domain anchoring protein. Each looping event triggers the simultaneous blocking of the two DNA binding sites (with DNA(site i) = 1 changed to DNA(site i) = 0), and each unlooping event leads to the simultaneous unblocking of the two DNA binding sites (with DNA(site i) = 0 changed to DNA(site i) = 1).

## Results

### RNAP Translocation-induced supercoiling mediates the collective behaviors of co-transcribing RNAP molecules

Kim *et al*. [13] found that"communication" between co-transcribing RNAP molecules on highly expressed genes manifests in two forms: cooperation and antagonism. Cooperation is the phenomenon that when the promoter is on (i.e., there is a continuous transcription initiation), each transcribing RNAP molecule elongates faster than the case when it moves solo. Antagonism is the phenomenon that when the promoter is off (i.e., no new transcription initiation), existing transcribing RNAP molecules elongate significantly slower than the case when the promoter is on. Kim *et. al*. proposed that these phenomena were due to the presence or absence of continuous loading of RNAP molecules onto the genes: the upstream negative supercoils produced by the last RNAP molecule could be rapidly canceled out by the downstream positive supercoils generated by a newly loaded RNAP molecule if the promoter is on, reducing the torsional stress, but remain unresolved if the promoter is turned off.

To explore whether the dynamics of transcription-induced supercoiling could indeed account fully for those collective behaviors, we simulated the transcription of *lacZYA* (the same systems used in Kim *et al*. [13]) centered in a 9 kb-long open domain, with both ends relaxing to the equilibrium supercoiling density -0.067 [3] rapidly (every 0.02 s). The rotation rate of RNAP was set at 0.2 $s^{-1}$, and the Topo I binding rate the same as the Gyrase binding rate. The supercoiling drift rate $k_{drift}$ is set as 50 $s^{-1}$, which yields a diffusion coefficient $D$ of 0.02 $\mu m^2 \cdot s^{-1}$. To represent promoters with different strengths, we varied the maximum initiation rate constant $k_{max}$ from 0.001 $s^{-1}$ to 0.2 $s^{-1}$, which is within the strength range of *E. coli* promoters [48]. Each simulation lasts for 1000 s and 1000 replicates are simulated. We calculated the empirical initiation rate and elongation rate for *lacZ* gene. The empirical initiation rate is calculated from the actual number of initiation events per unit time. The empirical elongation rate is calculated as the transcript length over the duration of each transcription. The elongation rate of a group of RNAP molecules is defined as the average elongation rate of this RNAP group. We only counted the initiation and elongation events from 750 s to 1000 s, since the mean mRNA copy number reaches a steady state after 750 s (**S3 Fig**).

**Fig 2A** shows that genes under higher initiation rates tend to have high elongation rates as well, recapitulating the cooperative behavior of RNAP molecules. Importantly, the cooperation is non-additive, i.e., the elongation rate"saturates" when the initiation rate is sufficiently high, which agrees with Kim *et al*.'s observations [13]. Consistent with this observation, the correlation in the displacement of adjacent RNAP molecules increased and plateaued at high initiation rates (**Figs 2B** and **S4**), indicating that RNAP tends to adjust its speed to mimic the speed of its neighbors, i.e., "cooperating" with each other. The distance between adjacent RNAP molecules continued to decrease in response to the increased initiation rate (**Fig 2C**). Interestingly, the average distance between two adjacent RNAP molecules was around 500 bp when the elongation rate reached the plateau and continued to decrease further in response to increased initiation rates, indicating that the full cooperation between adjacent RNAP molecules could operate over a distance of > 500 bp, which corresponds to an average number of 4 co-transcribing RNAP molecules at the same time (**S5 Fig**). The empirical initiation rate, empirical elongation rate, and number of co-transcribing RNAP molecules under different promoter strengths is listed in **S4 Table**. The distribution of elongation rate under different empirical initiation rates is shown in **S6 Fig**.

To determine how supercoiling contributes to the observed cooperation, we analyzed the supercoiling density and resisting torque distributions in the immediate vicinity of a transcribing RNAP molecule under the conditions of low, medium, and high empirical initiation rates

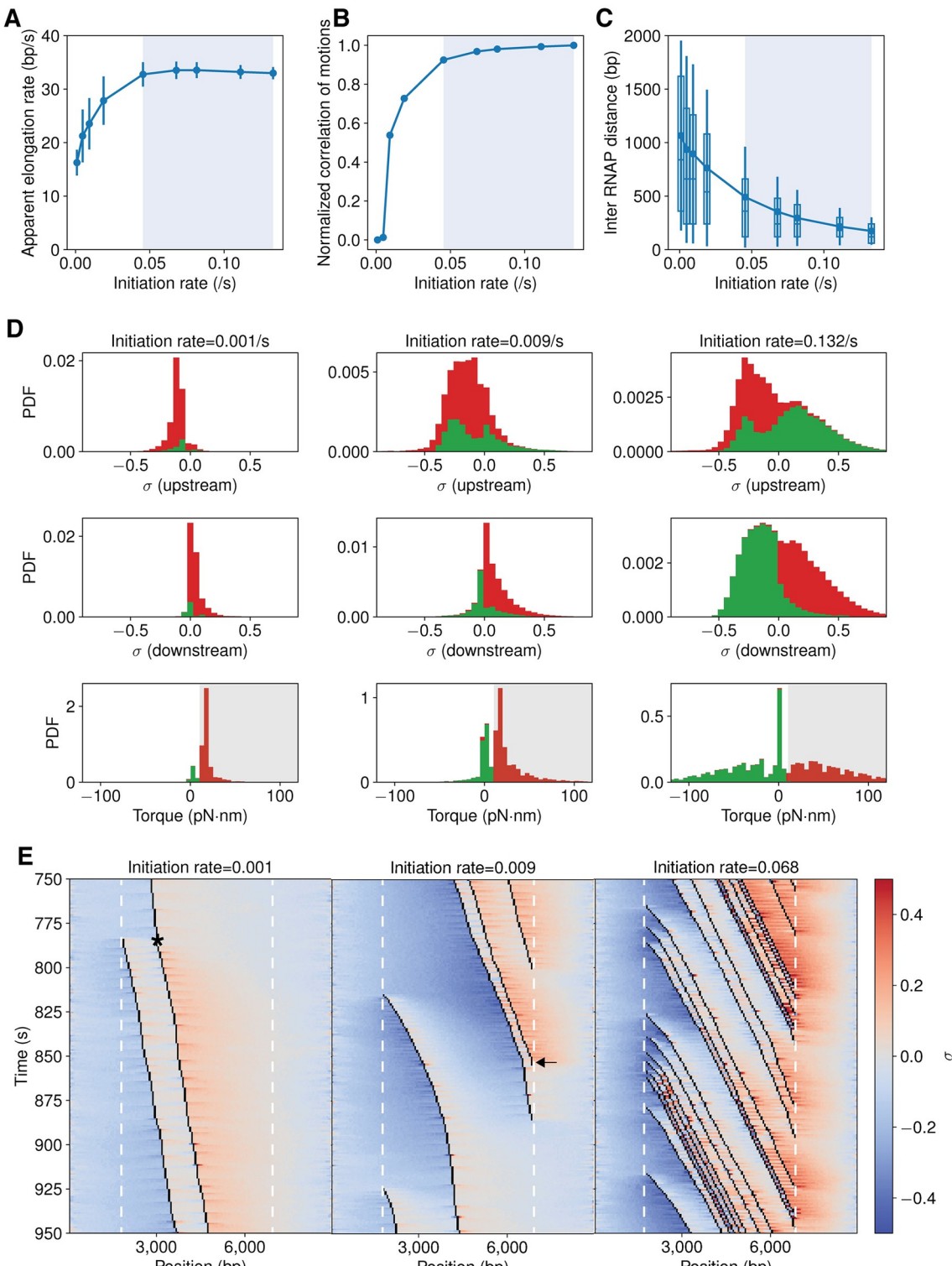

**Fig 2. Cooperation of RNAP transcription is favored by high initiation rates and antagonized by low initiation rates.** (A) The apparent RNAP elongation rate increases and reaches maxima (blue shaded region) as the initiation rate increases. Error bars are standard deviations from 1000 simulations. (B) Pearson correlation coefficient of adjacent RNAP trajectories increases and approaches unity (blue shaded region) as the initiation rate increases. The correlation coefficient is normalized such that "0" corresponds to the case of independently transcribing RNAP trajectories and "1" corresponds to the case of the co-transcribing RNAP trajectories in the highest

expressed gene. The unnormalized curve is in **S4 Fig**. (C) The mean distance (dots in boxes) between neighboring RNAP molecules decreases as the initiation rate increases. When the maximal cooperation is reached (intersect with blue shaded region, initiate rate at which maximal elongation rate is reached) the distance is ~ 500 bp. Box borders indicate first and third quartiles of the data, vertical lines standard deviations, and horizontal lines in the box media. (D) Stacked histograms of supercoil density upstream (top), downstream (middle) and torque (bottom) RNAP molecules face during transcription. Supercoil densities of processive and stalled RNAP molecules are shown in green and red, respectively. The torque value above the stall threshold is shaded in grey. (E) Representative kymographs of RNAP translocation trajectories (black lines) with the corresponding supercoiling density (positive in orange and negative in blue) in its vicinity. A time interval of 750–950 s was chosen to maintain a constant mean mRNA copy number at a steady-state under each condition (**S3 Fig**). The slope of the trajectory reflects the elongation speed of the associated RNAP molecule. Left and right white dashed lines indicate the start and end positions of the transcribing region, respectively. The asteroid in the left kymograph indicates that a slow-moving RNAP molecule started to move with a higher speed after a new RNAP molecule is loaded at the promoter on the left. The arrowhead in the middle kymograph indicates a transcribing RNAP molecule drastically reduces its elongation speed after the leading RNAP (right) terminates its transcription and dissociates.

(0.001, 0.019, and 0.131 $s^{-1}$) respectively (**Fig 2D**). At the low initiation rate, while the super-coiling densities upstream and downstream of a transcribing RNAP are relatively low (~ ± 0.1), they are not canceled out between adjacent RNAP molecules, because of the low loading rate of new RNAP molecules, and hence produce large positive torques around the RNAP molecule (**Fig 2D**, left column), decreasing the elongation rate and frequently stalling the RNAP molecule (red bars). As the initiation rate increases, the supercoiling densities around RNAP molecules also increase, but the corresponding torque decreases and becomes negative at the highest initiation rate (**Fig 2D**, middle and right columns) due to the cancellation of opposing supercoils between adjacent RNAP molecules. Consequently, majority of RNAP molecules under these conditions move processively with high elongation rates (green bars). Therefore, the reduced resisting torque can fully account for RNAP's cooperation in highly expressed genes. A full comparison of supercoiling density and torques under all initiation rate conditions is shown in **S7 Fig**. The distribution of stall duration is shown in **S8 Fig**.

To further illustrate the cooperative behavior of RNAP and its relationship with local super-coil density and torque, **Fig 2E** shows three exemplary kymographs of RNAP's translocation trajectories at the three different initiation rates. When the initiation rate is low (0.001 $s^{-1}$, **Fig 2E**, left), the overall supercoiling density of the DNA (negative upstream in blue and positive downstream in orange) is maintained at a relatively low level. However, these supercoils accumulate on both sides and generate high torques to stall the transcribing RNAP molecule frequently (vertical lines in trajectories). Interestingly, when a new RNAP molecule starts transcribing (left most trajectory at ~ 775 s in the simulation), the previously slow-moving RNAP molecule (indicated by *) started elongating faster than before, and its speed became similar to that of the new RNAP molecule, indicating cooperation. Under a high initiation rate (0.068 $s^{-1}$, **Fig 2E**, right), the overall supercoiling density on the DNA template is high, especially in the immediate upstream and downstream regions (dark blue or orange colors), but the torque that each RNAP molecule faces is low because the difference of the supercoiling density between the two sides of an RNAP molecule is small. Note that we also observed the antagonistic behavior of RNAP in these simulations, such as the one at ~ 850 s in the middle kymograph (initiation rate = 0.009 $s^{-1}$, **Fig 2E**, middle), in which when the leading RNAP mol-ecule finishes its transcription, the speed of the trailing one reduces drastically.

We also attempted to quantitatively reproduce the antagonistic behavior of RNAP mole-cules when the promoter is turned off as reported in Kim *et al.* [13]. We simulated the tran-scription of a strong promoter (with $k_{max}$ = 0.1 $s^{-1}$) and a weak promoter (with $k_{max}$ = 0.001 $s^{-1}$) and blocked the promoter activity after 2700 bp has been transcribed. Our simulation showed that RNAP molecules slowed down in response to promoter inactivation (**S9 Fig**), qualitatively recapitulating the antagonistic behavior Kim *et al.* have observed. However, the antagonistic behavior was observed on both promoters in our simulation, in contrast to what

was only observed on the strong promoter in Kim *et al*. [13]. It is possible that the way we simulated the promoter inhibition (by making the DNA segment of the promoter as a topological barrier and hence preventing further RNAP binding) introduced additional torsional stress than needed. A more careful treatment is needed in the future.

## The cooperative behavior requires fast Topo I unbinding and moderate supercoiling diffusion rates

We further investigate whether the cooperative behavior is sensitive to supercoiling diffusion rate and topoisomerase activities. We systematically varied the supercoiling diffusion coefficients from 0.002 to 2 $\mu m^2 \cdot s^{-1}$ and the Topo I unbinding rate from 0.001 to 100 $s^{-1}$ and simulated the transcription of various promoters (with the maximum initiation rate $k_{max}$ ranging from 0.001 $s^{-1}$ to 0.2 $s^{-1}$). To characterize the strength of the cooperation, we fit the empirical elongation rate—initiation rate curve with a piecewise linear function:

$$y = \begin{cases} k \cdot x + b, x < x_* \\ k \cdot x_* + b, x \geq x_* \end{cases}$$

with the slope $k$ of the initial rise of the elongation rate before it reaches a plateau at an initiation rate (denoted as $x^*$) indicating the strength of the cooperation (**Figs 3A–3C and S10**).

**Fig 3D** shows the phase diagram for the supercoiling diffusion coefficients and the Topo I unbinding rate. Dark blue squares indicate a large value of slope $k$, corresponding to strong cooperation. Strong cooperation is observed under moderate diffusion (D from 0.006 to 0.6 $\mu m^2 \cdot s^{-1}$) and fast Topo I unbinding ($k_{unbind}$ equal or greater than 0.1 $s^{-1}$). No cooperation is observed when D is larger than 0.6 $\mu m^2 \cdot s^{-1}$. **Fig 3B** shows a representative elongation rate—initiation rate curve for large D ($k_{unbind}$ = 1 $s^{-1}$, D = 2 $\mu m^2 \cdot s^{-1}$), where the elongation rate already achieves its maximum at low initiation rates. This observation is reasonable, since in this case, the diffusion of supercoiling is fast enough to dissipate the torsional stress, and it is no longer necessary to rely on opposing supercoiling to reduce torques between adjacent RNAP molecules. When Topo I unbinding is slow ($k_{unbind}$ = 0.01 $s^{-1}$, **Fig 3C**), the empirical initiation rates cannot reach a high value (0.05 $s^{-1}$), likely because the long dwell time of Topo I on the DNA allows it to remove excessive negative supercoils efficiently and hence inhibits transcription initiation. Overall, the above results show that RNAP cooperation can be achieved under a wide range of physiologically relevant conditions, indicating the robustness of this behavior to the perturbations of the physical properties of DNA and the binding affinity of topoisomerase.

To further validate our model, we next compared our model predictions with the experimental data in Kim *et al*. [13] by calculating the mean squared error (**Fig 3E**) $MSE = \frac{1}{n}\sum_{i=1}^{n}\left(Y_i - \hat{Y}_i\right)^2$, where $Y_i$ is the measured elongation rate from *lacZ* gene under three different IPTG induction levels by Kim *et al*. [13], and $\hat{Y}_i$ is the simulated elongation rate under the corresponding initiation rate. The parameterization of the initiation rate can be found in **S1 File**. The model agreed well with the experimental data (dark blue squares) within two regions of model parameters, one with D ranging from 0.02 to 0.06 $\mu m^2 \cdot s^{-1}$ and a Topo I unbinding rate greater than 0.1 $s^{-1}$ (black dashed box), and the other with D ranging from 0.2 to 0.6 $\mu m^2 \cdot s^{-1}$ and a Topo I unbinding rate at 0.01 $s^{-1}$ (grey dashed box). Although both regions of parameters agree with Kim *et al*. [13]'s data, the observation that DNA exists mostly in writhes (which displays slow diffusion) under physiological conditions [49] supports the first region. Therefore, we used D = 0.02 $\mu m^2 \cdot s^{-1}$ and Topo I unbinding rate = 1 $s^{-1}$ as parameters for the rest of the simulations in this study. Finally, we perturbed the RNAP

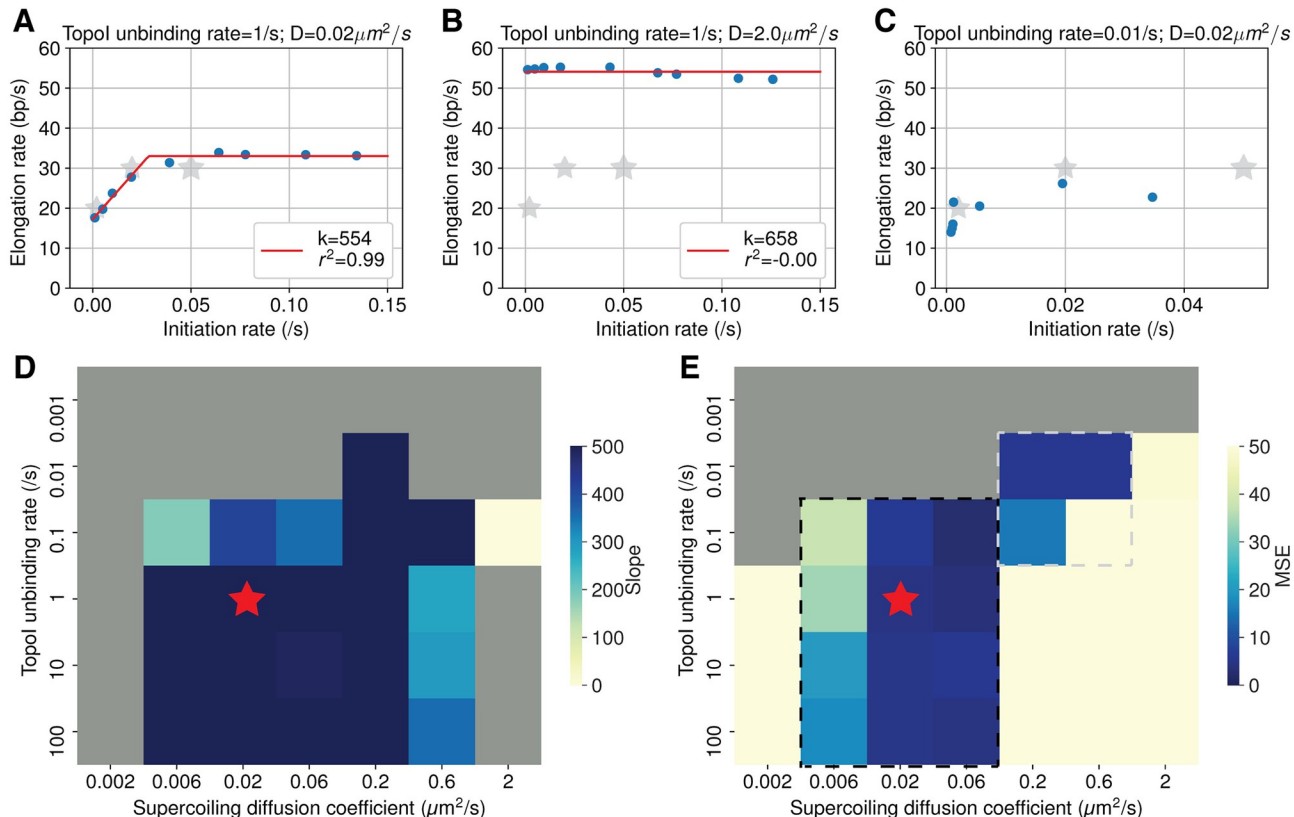

**Fig 3. Cooperation holds under the condition of fast Topo I unbinding and moderate supercoiling diffusion. (A to C)** Examples of the elongation rate–initiation rate curves (blue dots) and the corresponding piecewise linear fitting (red lines) for three different combinations of the Topo I unbinding rate and supercoil diffusion rate. Fitting is only performed for scenarios where the maximum empirical initiation rate is greater than 0.05 $s^{-1}$ (i.e., **C** is not fitted). The fitted slope ($k$) values and fitting goodness ($r^2$) were displayed in the legend. Grey stars correspond to the experimental data in Kim *et al.* [13]. **(D)** Heatmap of RNAP cooperativity, characterized by the slope of the elongation rate–initiation rate curve, under different supercoiling diffusion coefficients and Topo I unbinding rates. The grey region corresponds to the regime where the fitting goodness $r^2$ is below 0.85, or the fitting is not performed (maximum initiation rate < 0.05 $s^{-1}$). The red star indicates parameters used in the rest of the paper. **(E)** Heatmap of the mean squared error (MSE) between the model and the experimental data from Kim *et al.* [13].

counterrotation rate locally around the chosen parameters (D = 0.02 $\mu m^2 \cdot s^{-1}$, Topo I unbinding rate = 1 $s^{-1}$, **S11 Fig**). We found that as long as RNAP counterrotates slowly ($k_{rot} <= 1$ $s^{-1}$), the cooperation remains, suggesting that the cooperative behavior is not sensitive to our specific choice of RNAP counterrotation rate.

## Supercoiling accumulated in a topological domain modulates transcriptional noise

Bacterial transcription exhibits a large cell-to-cell variability [42,50,51,52], allowing a population of isogenic cells to exhibit heterogenous phenotypes and responses to various stimuli [53]. The variability comes from either the stochastic nature of molecular reactions (intrinsic noise) or the variation in molecule copy numbers in different cells (extrinsic noise) [54]. Here we are interested in understanding how supercoils accumulated in a topological domain could function as an additional factor to contribute to the intrinsic noise in the mRNA production of a gene. We characterize the transcriptional noise by the Fano factor of mRNA copy number, which equals one when mRNA's production is a stochastic, Poissonian process of random birth and death [50]. When mRNAs are produced in bursts due to random activation and

inactivation as observed in both prokaryotes and eukaryotes, the Fano factor is greater than one and reflects the mRNA burst size [55].

We first simulated a 4.2 kb open DNA with a 2.4 kb gene located in the center, with the ends constrained and relaxing every 0.02 s. Each condition is simulated for 2000s with 1000 replicates. As shown in **Fig 4A, 4B, 4D and 4E**, under this condition both weak and strong promoters yield an mRNA copy number distribution with a Fano factor around 1 due to the stochastic, Poissonian transcription. When we introduce a topological barrier at 0.06 kb and 4.14 kb, which dynamically loops (for 5 min) and unloops (for 1 min) the intervening DNA (to mimic realistic chromosomal DNA dynamics), both the weak and strong promoters show a substantial drop in the mean mRNA production (**Fig 4C, 4F, 4G and 4H**). However, the strong promoter exhibits a larger Fano factor in mRNA copy number distribution than the weak promoter, suggesting that additional transcription noise is introduced (**Fig 4C and 4F**). The supercoiling density in the dynamically looping DNA exhibits strong positive or negative values, whereas that of the open DNA maintains a moderate and relatively constant level (**Fig 4G and 4H**, bottom panel). Notably, the transcription initiation of the dynamically looping DNA is highly bursty, and each burst corresponds to the release of accumulated of torsional stress when the loop opens (**Fig 4H**, bottom panel, red shaded region). The elongation dynamics also contribute to the noise (**S12 Fig**), where the elongation rate (**S12A Fig**) and number of co-transcribing RNAP (**S12B Fig**) displays greater variability in the dynamically looping case than in the unlooped case.

As in all our simulations we also included the actions of Topo I and Gyrase in these simulations. However, we observed that topoisomerases cannot effectively alleviate the torsional stress generated by genes with strong promoters in a topologically constrained DNA loop. The stress is largely released only when the DNA loop opens. Consistent with this expectation, genes with strong promoters are transcribed with larger noises under the dynamically looping condition, and the Fano factor increases with the promoter strength (**Fig 4I**). Additionally, when the loop is closed, the torsional stress reduces the transcription initiation rate. In contrast, when the loop is open, the torsional stress is released and transcription initiates with high rates (**Fig 4J**). These observations further support the notion that the difference in transcription activities during the looped and unlooped states underlies bursty transcription and contributes to additional transcriptional noise in mRNA production.

We note that previous works [21,25] attributed transcription bursting to the binding and unbinding of Gyrase in a topological domain. This mechanism is based on the assumption that Gyrase's binding is a rare event compared to the timescale of transcription. Our model suggests that even if the binding and unbinding of Gyrase are far more frequent than transcription initiation, transcription can still be bursty due to the topological domain formation and opening. Furthermore, the timescale of domain formation and dissolution plays a significant role in bursty transcription: the more frequent the DNA switches between looped and unlooped state, the less bursty transcription is. We show that when the dwell time of looping is reduced from 5 min to 1 min, the Fano factor drops accordingly (**S13 Fig**, green to orange curve); if the DNA is always looped, the transcription will stay in a nearly dormant state as the torsional stress cannot be released effectively even in the presence of topoisomerase activities; consequently, the resulting Fano factor is down to unity (**S13 Fig**, red curve). Note that our model does not exclude the possibility of transcription bursting resulting from strong topoisomerase binding/unbinding, which is rare [56], but instead provides a mechanism that can result in transcription bursting independent of topoisomerase activities.

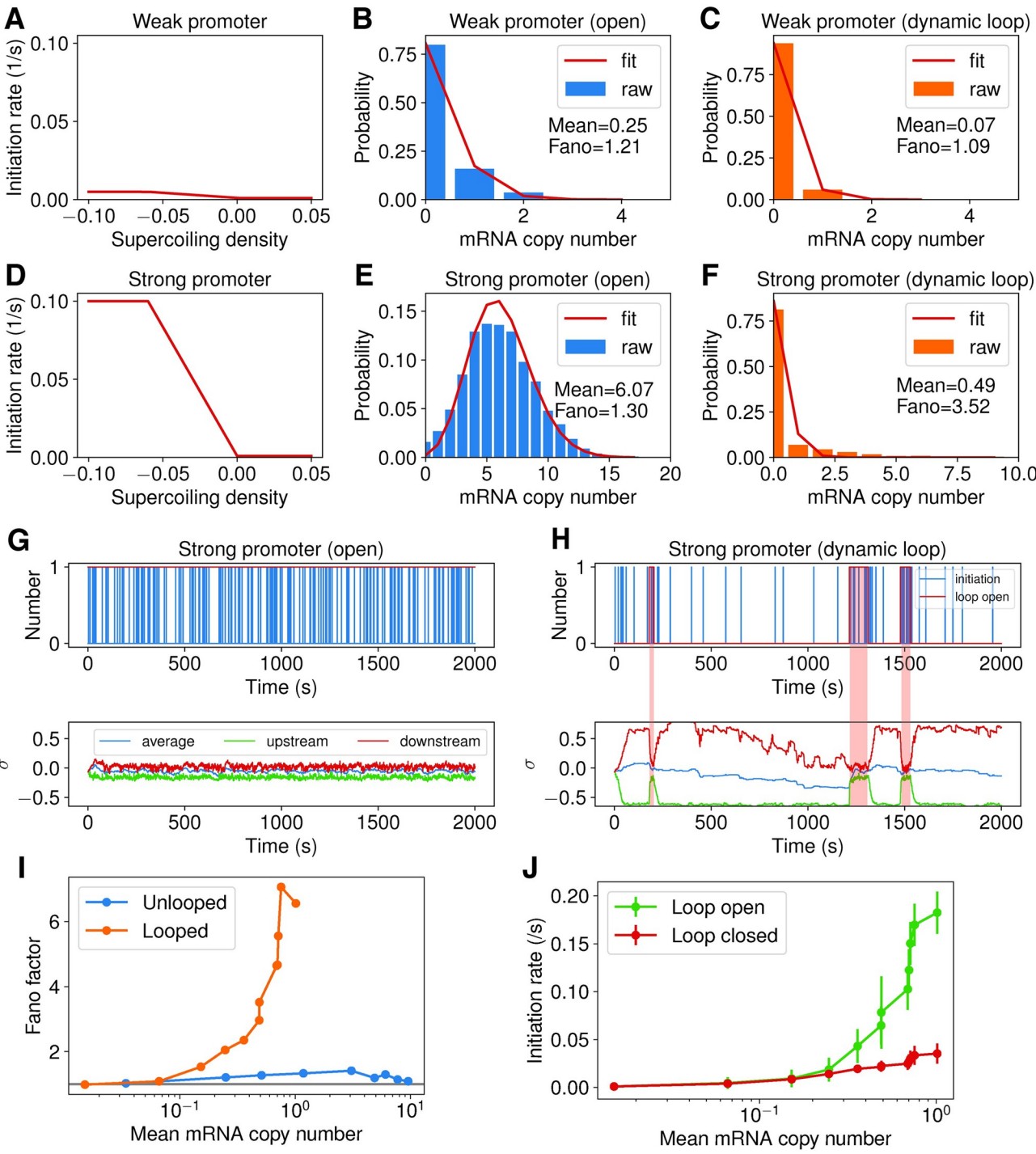

**Fig 4. Dynamic topological domain formation results in bursty and noisy transcription from strong promoters through the accumulation and release of supercoiling.** (A-F) Comparison of a weak promoter ($k_{max}$ = 0.005 $s^{-1}$, top panel) and a strong promoter ($k_{max}$ = 0.1 $s^{-1}$, bottom panel) in their responses to supercoil density (A and D), mRNA copy number distribution when the DNA is open (unlooped, B and E) and when the DNA dynamically loops and unloops (C and F). (G-H) Exemplary mRNA production time traces (top panel) and the corresponding supercoil density of the DNA (bottom panel) for the strong promoter in the open-DNA condition (G) and the dynamically looping condition (H). The pink-shaded regions in H indicate the time when the loop is open. (I) Comparison of the Fano factor as a function of mean mRNA copy number for genes in an open DNA (blue) and that in a dynamically looping DNA condition (orange). The grey line denotes Fano factor = 1. (J) Comparison of the average initiation rate of genes when the DNA is open (green) and when the DNA is closed (red) in the dynamically looping DNA condition.

## Intergenic supercoiling mediates communication between two neighboring genes

Previous works show that neighboring genes could influence the transcription of each other, and the interaction is dependent on the relative orientation of the genes [15,16,57]. It was proposed that intergenic supercoiling dynamics may mediate communications between neighboring genes [13,15,57]. To investigate this hypothesis, we simulated two genes of the same length (1.2 kb) and promoter strength ($k_{max}$ = 0.05 $s^{-1}$) but arranged differently (convergent, divergent or codirectional) in a 9.6-kb long open DNA. To examine the effects of supercoiling on transcriptional activities, we also simulated the cases where supercoiling is eliminated by increased topoisomerase activities. Specifically, the Topo I and Gyrase binding rates are increased by 50 times relative to normal conditions, and the catalytic rates are increased by 1000 times. Each condition is simulated for 2000s with 1000 replicates.

We first analyzed the effects of intergenic supercoiling on mRNA production with the intergenic distance held at 1.2 kb (**Fig 5A**). For convergent and divergent arrangements, since the two genes displayed the same transcription activities due to the symmetry, we only showed the mRNA distribution for the upstream gene. For codirectional genes, we examined the mRNA distribution for both the upstream (trailing) and the downstream (leading) genes. As shown in **Fig 5B**, we observed that the presence of supercoils led to a significant decrease in the mean mRNA production of the genes in the convergent arrangement (leftmost panel, orange bars) but no change to the genes in the divergent arrangement (middle panel, orange bars) compared to the condition when supercoils on the DNA are eliminated efficiently by topoisomerase activities (blue bars). For the upstream gene in the codirectional arrangement (second panel from the right), mRNA production exhibited a significant increase in the presence of supercoiling. For the downstream gene in the codirectional arrangement, we observed a significant decrease in mRNA production (rightmost panel). These results demonstrate that supercoiling indeed influences the transcription activity of genes in different orientations differentially.

To examine how supercoiling contributes to the above observations, we plotted intergenic supercoil density distributions of these genes in **Fig 5C**. For convergent genes (left panel), the intergenic region is dominated by positive supercoils, which are produced by the transcription-induced supercoiling downstream of both genes, and hence inhibiting the transcription of both genes. In contrast, the region between divergent genes is dominated by negative supercoils (middle panel), which are produced upstream by the promoters of both genes. This hyper negative supercoiling led to increased transcription initiation (**Fig 5D**) but decreased transcription elongation (**Fig 5E**), resulting in a modest decrease in mRNA production. For codirectional genes, the intergenic region is also dominated by positive supercoiling (right panel), although the densities are lower compared to that of convergent genes. This is an interesting observation, because intuitively one would expect that positive supercoils produced by the upstream gene and the negative supercoils produced by the downstream gene would cancel each other, leading to relaxed intergenic DNA. In reality, however, the upstream gene will roughly maintain its transcription activity over time (**Fig 5D**) since its promoter supercoiling is frequently reset by the default negative supercoiling from the chromosome end. On the other hand, the downstream gene will experience a decline in transcription due to the immediate suppression of transcription initiation by the relaxed intergenic DNA. This difference gives rise to a positive feedback loop where the increased positive supercoiling produced by the upstream gene further decreases the initiation rates of the downstream gene. Consequently, the upstream gene does not change its mRNA production level dramatically, but the downstream gene decreases its mRNA production level.

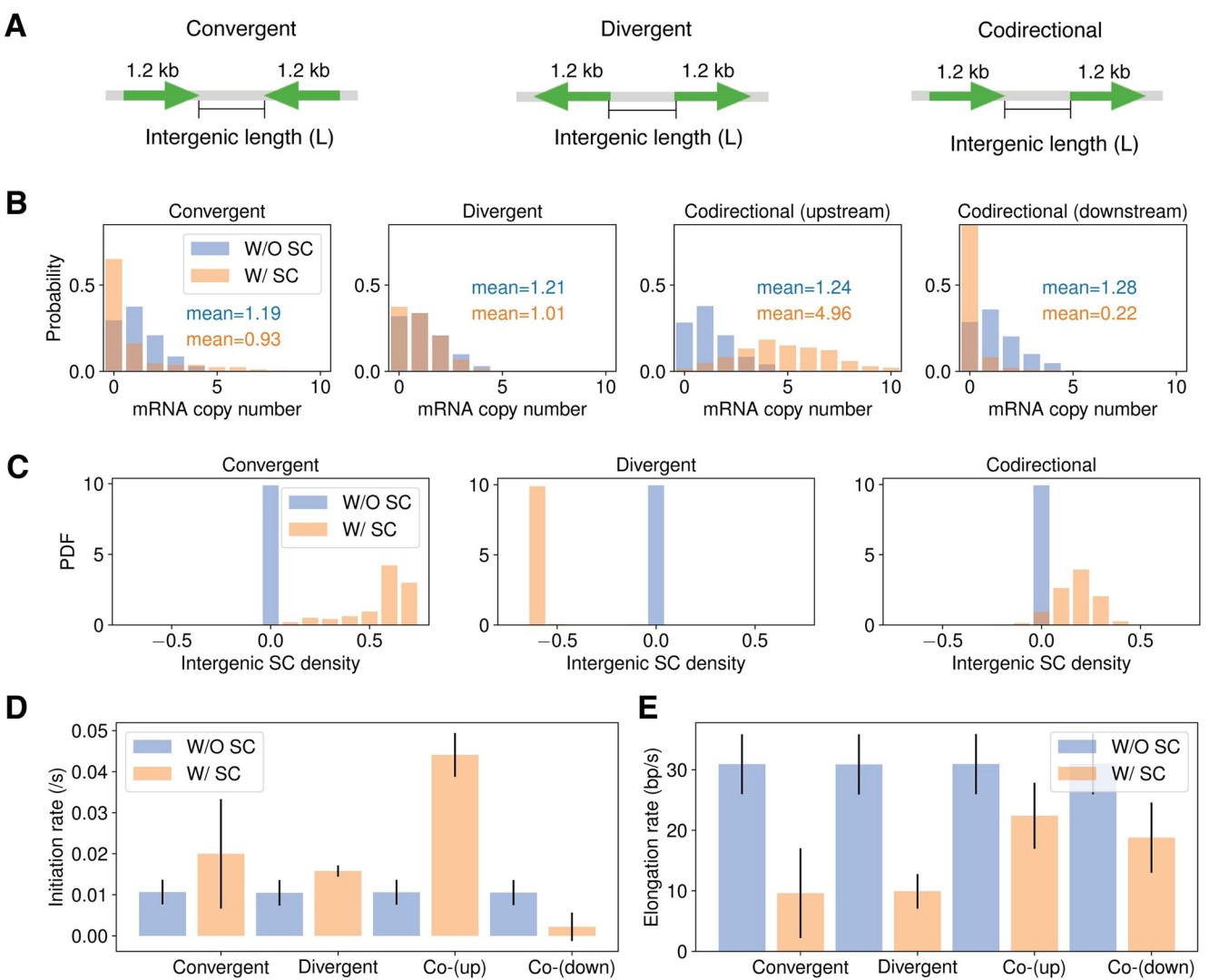

**Fig 5. Intergenic supercoiling affects the level of mRNA production of two neighboring genes.** (A) The construct for convergently, divergently and codirectionally arranged genes with the intergenic length fixed at 1.2 kb. The mRNA copy number distribution (B) and intergenic supercoiling distribution (C) for cases with (orange bar) and without (blue bar) intergenic supercoiling. The bar plot for initiation rate (D) and elongation rate (E) for cases with (orange bar) and without (blue bar) intergenic supercoiling. Error bar suggests standard deviation.

Based on the above results, we reasoned that the presence of supercoiling should also render mRNA production of one gene sensitive to changes in the initiation rate of the other adjacent gene. To examine this hypothesis, we varied the initiation rate of one gene (**Fig 6A**) and monitored the resulting mRNA production levels of the other gene (**Fig 6B**), the intergenic supercoiling density (**Fig 6C**), and the elongation rate the other gene (**Fig 6D**) for the three gene arrangements.

For convergent genes (leftmost, **Fig 6B**), as we increased the initiation rates of one gene, the mean mRNA copy number of the other gene decreases gradually. This result is likely because when the initiation rate of the first gene is high ($k_{max} > = 0.02\ s^{-1}$), the intergenic region become significantly positively supercoiled (leftmost, **Fig 6C**), inhibiting the transcription of the other gene. For divergent genes (second from the left, **Fig 6B**), we also observed that mRNA production decreases with the initiation rate of the neighboring gene, likely due to the

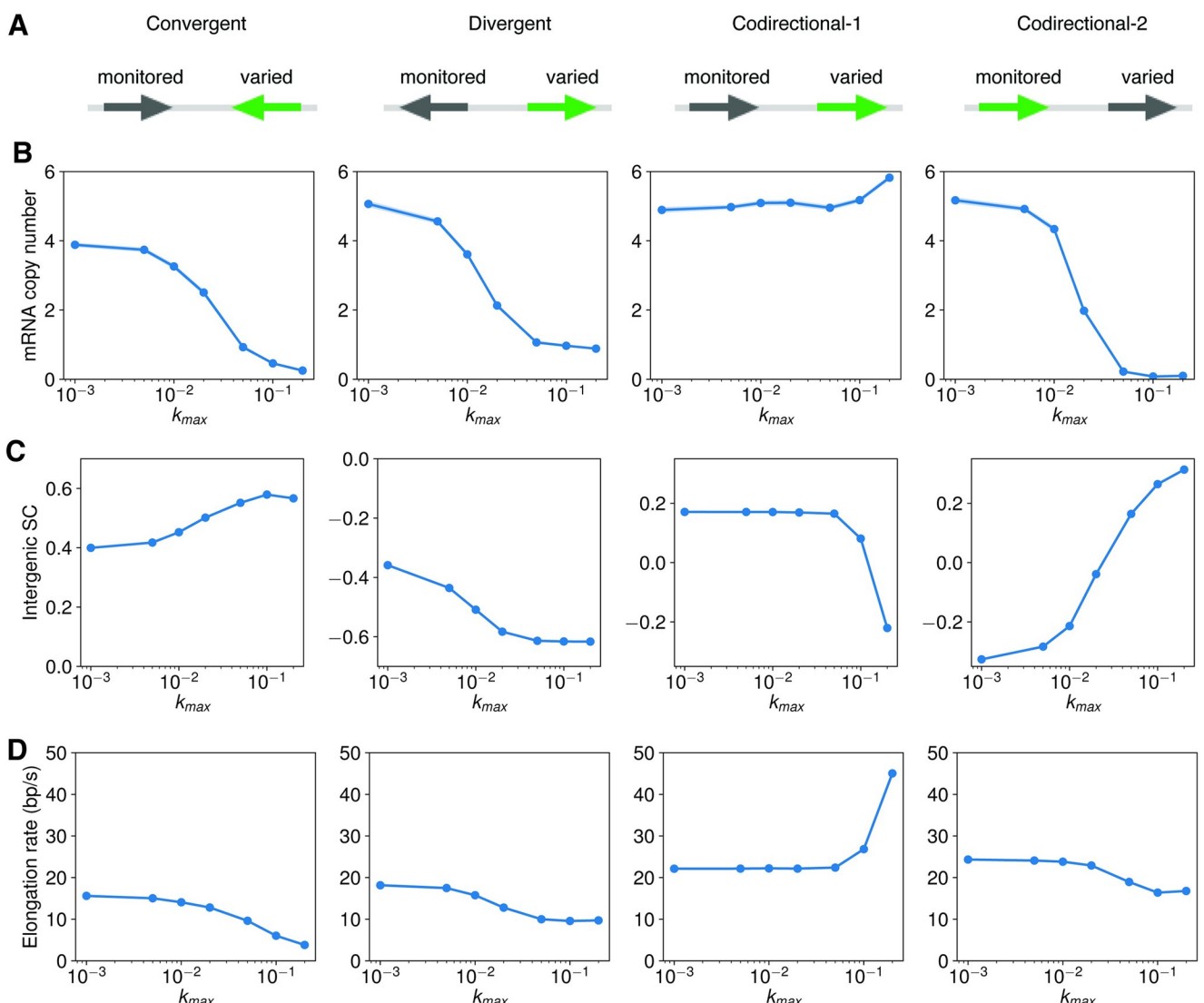

**Fig 6. The effects of the initiation rate on the expression of adjacent genes.** (A) The construct for convergently, divergently, and codirectionally arranged genes where the initiation rate of one gene is varied ($k_{max}$ = 0.001, 0.005, 0.01, 0.02, 0.05, 0.1, 0.2 $s^{-1}$) and the initiation rate of the other gene is monitored in response to that of the first gene with an upper limit of $k_{max}$ = 0.05 $s^{-1}$. (B) The mean mRNA copy number of the monitored gene as a function of the maximum initiation rate ($k_{max}$) of the first gene, for the four constructs shown in (A). The dot is the mean and the shaded area is mean ± SEM. (C) The intergenic supercoiling density as a function the maximum initiation rate ($k_{max}$) of the adjacent gene. The dot is the mean and shaded area is mean ± SEM. (D) The empirical elongation rate of the monitored gene as a function of the maximum initiation rate ($k_{max}$) of the gene whose initiation rate is varied. The elongation rate is calculated for transcription events after 750 s, where the mean mRNA copy number already reaches a steady state (**S14 Fig**).

hyper-negative intergenic supercoiling density (second from the left, **Fig 6C**) and the resulting diminished elongation rate (from 18 bp/s to 10 bp/s, second from the left, **Fig 6D**). The decreased elongation rate is qualitatively consistent with the observation in Kim *et al.* [13], where the elongation rate of *lacZ* decreases from ~30 bp/s to ~ 20 bp/s as its neighboring divergently oriented gene increases its promoter strength. For codirectional genes, since the gene arrangement is asymmetric, we modulated the initiation rates for both the upstream gene and the downstream gene separately. When the initiation rate of the downstream gene is increased, we observed no change in the mRNA production of the upstream gene (third from the left, **Fig 6B**) and no change in the intergenic supercoiling density (third from the left, **Fig 6C**) until a

very high initiation rate ($k_{max}$ = 0.2 $s^{-1}$) of the downstream gene is reached. The homeostasis in intergenic supercoiling density suggests a feedback loop: the more the downstream gene transcribes, the more negative supercoiling is generated in the intergenic region, and the more the upstream gene is activated, which produces more positive supercoils in the intergenic region to annihilate the negative supercoils. Conversely, when the initiation rate of the upstream gene is increased, we observed a decrease in the expression of the downstream gene (rightmost, **Fig 6B**), accompanied with an increased intergenic supercoiling density (rightmost, **Fig 6C**).

## Discussion

In this study, we modeled supercoiling-dependent transcription systems in *E. coli*. This model contains the following unique features. First, we provided an explicit description of supercoiling-sensitive elongation by modeling the stall propensity to be torque-dependent. Ma *et al.* [10] observed that torque could regulate both RNAP's stall propensity and the velocity for processive RNAP. However, the velocity's dependence on torque is relatively weak (for example, the velocity only drops from about 24 *bp/s* to about 18 *bp/s* when the torque changes from −6 *pN·nm* to 7.5 *pN·nm*). Hence, we argue that the elongation rate is mostly regulated by the stall propensity, and the actual change in the velocity is negligible. Second, our model is pertinent to the realistic biological context. We modeled the molecular events of binding/unbinding of topoisomerases, including Topo I and Gyrase, formation/dissociation of topological domains, translocation and counterrotation of RNAP, and diffusion of supercoils along the DNA. Integrating all these biological events that occur in live cells with physiologically relevant parameters renders the model explicit in its biological implications.

Using this model, we quantitatively reproduced the orchestration between co-transcribing RNAP molecules over long distances [13]. We showed that the cooperation between RNAP molecules, manifested by the enhancement in elongation rate in highly expressed genes, is mediated by supercoiling cancellation and the consequently reduced torque generated by RNAP. It is noteworthy that previous models [20,26,28] treat the supercoiling diffusion between topological barriers as an infinitely fast process. However, by explicitly modeling supercoiling diffusion, we showed that the actual diffusion coefficient matters. The cooperative behavior between RNAP molecules is only achieved when supercoiling diffusion is within a reasonable range (D from 0.006 to 0.6 $\mu m^2 \cdot s^{-1}$). Since no comprehensive measurement for DNA supercoiling diffusion exists so far, we call for future experiments to test this prediction.

We then utilized this model to explore the role of promoter strength and the formation of topological domain on transcription regulation mediated by supercoiling. Our simulations showed that a dynamically looped topological domain could reduce the average mRNA production within the domain and add intrinsic noise to gene transcription in a promoter strength-dependent way. Thus, topological domains could act as transcription regulators and globally manipulate the distribution of transcription levels. We also provided another mechanism for transcriptional bursting by the dynamic formation and dissolution of topological domains, which is in addition to Gyrase binding/unbinding as previously proposed [21,25]. Note that the necessary condition for supercoiling-mediated transcriptional bursting is that there is enough timescale separation in supercoiling generation and dissipation. However, the timescale Gyrase binding/unbinding is still controversial: the *in vitro* study by Chong *et al.* [8] estimated the Gyrase to bind/unbind every several minutes, while the *in vivo* experiment by Stracy *et al.* [30] observed each binding to span only 2–10 seconds. If the latter picture is true, Gyrase activity alone will not provide enough time scale separation for supercoiling generation and dissipation. On the other hand, we also lack the experimental evidence for topological

domain-induced transcriptional bursting: although the stability of several transcription factor-mediated loops has been measured [58,59,60] (such as LacI- or CI-mediated loop, whose stability ranges from seconds to minutes), the kinetics of other kinds of topological domains (formed by nonspecific nucleoid-associated proteins like H-NS [61]) remain largely unknown. More work is needed to elucidate the mechanism underlying transcriptional bursting.

We also analyzed the modulation of mRNA production of one gene by a neighboring gene and its dependence on the relative orientation of the two genes. We showed that even a simple two-gene system could yield complex behaviors due to supercoiling dynamics (for example, the asymmetry in gene regulation in codirectional genes). These behaviors brought about the need to use supercoiling-dependent models in facilitating the design of synthetic genetic circuits [62].

One caveat in this work is that we did not model the explicit random walk of supercoiling due to the associated computational cost; instead, we modeled a biased random walk where the directionality is deterministic and the waiting time between movements is stochastic. In the former case, the stochastic movement of supercoiling might trigger extra responses of RNAP molecules and alter their behavior. To examine whether RNAP's transcription activity behaves differently under these two assumptions, we simulated both models in the same system used in **Fig 2**, with Topo I unbinding rate fixed ($1 \text{ s}^{-1}$) and supercoiling diffusion coefficient varied (0.002, 0.02, and 0.2 $\mu m^2 \cdot s^{-1}$). We cannot simulate a higher diffusion coefficient with the explicit random walk model due to computational limitations. We found no major difference between the two models in the response of mean mRNA production to promoter strength (**S15A Fig**), and the response of empirical elongation rate to empirical initiation rate (**S15B Fig**). The response of transcriptional noise to mRNA production looks slightly different (**S15C Fig**), but this might be due to the small sample size used in our simulation and needs further investigation.

We also need to point out that our model neglected the complex dynamics of supercoiling, i.e., the different diffusion behavior of the twist and the writhe [63] and the hopping of plectonemes [29]. According to a recent two-phage dynamic model of supercoiling [63], twist diffusion reaches fast equilibrium, compared to the RNAP translocation, while writhe diffusion is slow. Meanwhile, the transition from the twisted to the writhed state could happen dynamically during the translocation process [63]. However, our model did not differentiate the twisted or writhed states and used a single diffusion coefficient to describe the transport of the linking number. Furthermore, according to van Loenhout *et al.* [29], plectonemes could suddenly disappear and reappear at thousands of base pairs away on the DNA, known as "hopping." The hopping behavior does not rely on diffusion. It is worth investigating how those complex behaviors affect transcription with a more refined physical model.

Another potential concern of our model is that we assumed a constant RNAP rotation rate and did not let the rotational drag of RNAP grow as the transcript gets longer, contrasting with the treatment in previous models [20, 26, 28]. When the length-dependent rotational drag of RNAP is considered, the RNAP would be harder to rotate as the transcript becomes longer, and more complicated behavior might be yielded. If the RNAP rotation is the primary pathway to release the inter-RNAP torsional stress, for very strong promoters, the elongation rate will be limited by the rotation velocity, and the elongation rate could decrease as the initiation increases. This phenomenon will be more pronounced in longer genes. The non-monotonic variation of elongation rate with promoter strength and its gene length dependence is a unique behavior of the length-dependent rotational drag model [26]. Although our model cannot make that kind of prediction, it is a good approximation when RNAP rotation is not the dominant pathway for inter-RNAP torsional stress release. Our model shows that as long as the RNAP molecules rotate slowly enough (i.e., the rotational drag is large enough), the

cooperative behavior between RNAP molecules holds and is not sensitive to the specific rotation rate (**S11 Fig**), which qualitatively agrees with the arguments in [28].

## Supporting information

**S1 Fig. Schematic of torsional DNA buckling.** The left end of DNA is fixed on a topological barrier, and the right end is pulled by force F. Initially, DNA exists in the form of pure twists. We reduce the linking number by one at each step (and it could be removed by topoisomerase or other mechanisms). At first, DNA reduces the density of twists. When the linking number is reduced from 9 to 8, the DNA buckles itself to form a negative supercoil and preserves the number of twists in the previous step.
(PDF)

**S2 Fig. The generation and release of torsional stress by RNAP during transcription.** (A) The displacement of RNAP generates negative supercoiling in the upstream region and positive supercoiling in the downstream region. The build-up of these supercoiling could stall a transcribing RNAP molecule. (B) The counterrotation of RNAP (and the associated DNA) could return the DNA to a relaxed state without changing the total linking number of the system. The effect is equivalent to the diffusion of one downstream positive twist to the upstream negative twist to annihilate it.
(PDF)

**S3 Fig. The unnormalized Pearson correlation coefficient of adjacent RNAP trajectories versus the initiation rate.** For each condition, 1000 replicates were simulated. Shaded area corresponds to the initiation rates where the apparent elongation rate reaches the maxima.
(PDF)

**S4 Fig. Mean mRNA production over time under different empirical initiation rates.** For each condition, 1000 replicates were simulated. From the figures, we could roughly tell that the mean mRNA copy number stabilizes after about 750 s, suggesting that the system reaches a steady state.
(PDF)

**S5 Fig. Empirical elongation rate as a function as the number of co-transcribing RNAP molecules.** The number of co-transcribing RNAP molecules is only calculated when there is active transcription. The dot is the mean and shaded area is mean ± SD.
(PDF)

**S6 Fig. The histogram of elongation rate under different empirical initiation rates.** For each condition, 1000 replicates were simulated. The elongation rate is calculated as the transcript length over the duration of the transcription.
(PDF)

**S7 Fig. The stacked histogram of immediate upstream and downstream supercoiling density and torques of RNAPs under different empirical initiation rates.** Two populations are shown: processive RNAP (green) and stalled RNAP (red). For each condition, 1000 replicates were simulated. The torque value above the stall threshold is shaded in grey.
(PDF)

**S8 Fig. The histogram of RNAP stall duration under different empirical initiation rates.** For each condition, 1000 replicates were simulated.
(PDF)

**S9 Fig. The effect of promoter inactivation on elongation rate.** The transcription of a group of RNAP molecules corresponds to a $k_{max}$ of 0.1 $s^{-1}$. The transcription of a solo RNAP molecule corresponds to a $k_{max}$ of 0.001 $s^{-1}$. Transcription without promoter inactivation is denoted as "On" (grey color). Transcription with promoter inactivation at 2700 bp is denoted as "Off" (purple color). Error bar suggests standard deviation.
(PDF)

**S10 Fig. The empirical elongation rate–initiation rate curve for all conditions.** The Topo I unbinding rate is varied from 0.001 to 100 $s^{-1}$ (row). The supercoiling diffusion rate is varied from 0.002 to 2 $\mu m^2 \cdot s^{-1}$ (column). For $k_{max}$ = 0.001 s$^{-1}$, 500 replicates are simulated. For $k_{max}$ = 0.005, 0.01, 0.02 s$^{-1}$, 100 replicated are simulated. For $k_{max}$ = 0.05, 0.08, 0.1, 0.15, 0.2 s$^{-1}$, 10 replicates are simulated. The blue dot corresponds to simulated data, and the red curve corresponds to the piece-wise linear fitting. The fitted slope k and R squared value is shown. The grey star corresponds to the experimental data in Kim *et al*. [1].
(PDF)

**S11 Fig. The empirical elongation rate–initiation rate under different RNAP counterrotation rates.** Error bar suggests standard deviation. For $k_{max}$ = 0.001 s$^{-1}$, 500 replicates are simulated. For $k_{max}$ = 0.005, 0.01, 0.02 s$^{-1}$, 100 replicated are simulated. For $k_{max}$ = 0.05, 0.08, 0.1, 0.15, 0.2 s$^{-1}$, 10 replicates are simulated.
(PDF)

**S12 Fig. The distribution for empirical initiation rate, empirical elongation rate, and number of co-transcribing RNAP molecules for a strong promoter with $k_{max}$ = 0.1 $s^{-1}$.** The distribution is drawn from 1000 simulations. Orange bar corresponds to the dynamically looping cased, and blue bar corresponds to the unlooped case.
(PDF)

**S13 Fig. The Fano factor as a function of mean mRNA copy number under different conditions.** Open (blue): open chromosome end (relaxing to equilibrium every 0.2 s). Dynamic1 (orange): dynamic looped (on average looping for 1 min and unlooping for 1 min). Dynamic2 (green): (on average looping for 5 min and unlooping for 1 min). Closed (red): permanent loop. 1000 replicates are simulated.
(PDF)

**S14 Fig. The mean mRNA production over time for the monitored gene in the two-gene system.** The curve is averaged from 1000 simulations.
(PDF)

**S15 Fig. Comparison between explicit random walk and biased walk model.** (A) Mean mRNA production as a function of $k_{max}$ for the biased random walk model (orange) and the explicit random walk model (blue). For $k_{max}$ = 0.001 s$^{-1}$, 500 replicates are simulated. For $k_{max}$ = 0.005, 0.01, 0.02 s$^{-1}$, 100 replicated are simulated. For $k_{max}$ = 0.05, 0.08, 0.1, 0.15, 0.2 s$^{-1}$, 100 replicated are simulated for D = 0.002 $\mu m^2 \cdot s^{-1}$, and 10 replicates are simulated for D = 0.02 and 0.2 $\mu m^2 \cdot s^{-1}$. (B) Empirical elongation rate as a function of empirical initiation rate. (C) Fano factor as a function of mean mRNA copy number. The dot is the mean, and the shaded area is mean ± SEM.
(PDF)

**S1 Table. Species index.**
(DOCX)

**S2 Table. Model equations.**
(DOCX)

**S3 Table. Parameter estimation.**
(DOCX)

**S4 Table. Empirical initiation rate, elongation rate and RNAP density under different promoter strengths.**
(DOCX)

**S1 File. Supplementary Notes.**
(DOCX)

## Acknowledgments

We would like to thank Dr. Taekjip Ha for useful comments on the initial draft.

## Author Contributions

**Conceptualization:** Yuncong Geng, Christopher Herrick Bohrer, Elijah Roberts.

**Formal analysis:** Yuncong Geng.

**Funding acquisition:** Jie Xiao, Elijah Roberts.

**Investigation:** Yuncong Geng.

**Methodology:** Yuncong Geng.

**Resources:** Elijah Roberts.

**Software:** Yuncong Geng, Elijah Roberts.

**Supervision:** Jie Xiao, Elijah Roberts.

**Visualization:** Yuncong Geng.

**Writing – original draft:** Yuncong Geng.

**Writing – review & editing:** Yuncong Geng, Christopher Herrick Bohrer, Nicolás Yehya, Hunter Hendrix, Lior Shachaf, Jian Liu, Jie Xiao.

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
