## [Decision Letter · Decision Letter 0]

23 Feb 2022

Dear Ms Geng,

Thank you very much for submitting your manuscript "A spatial stochastic model reveals the role of supercoiling in transcription regulation" for consideration at PLOS Computational Biology.

As with all papers reviewed by the journal, your manuscript was reviewed by members of the editorial board and by several independent reviewers. In light of the reviews (below this email), we would like to invite the resubmission of a significantly-revised version that takes into account the reviewers' comments.

Specifically, please revise the manuscript following the reviewers' comments to improve the clarity of the model descriptions, result interpretations, and discussions.

We cannot make any decision about publication until we have seen the revised manuscript and your response to the reviewers' comments. Your revised manuscript is also likely to be sent to reviewers for further evaluation.

Sincerely,

Chongzhi Zang

Guest Editor

PLOS Computational Biology

Ilya Ioshikhes

Deputy Editor

PLOS Computational Biology

Reviewer's Responses to Questions

**Comments to the Authors:**

Reviewer #1: In general, this is a good paper which addresses an important and timely issue related to the role of supercoiling in modulating transcriptional dynamics. The authors have mostly done a good job in describing their ideas and in showing how they lead to a variety of interesting biological findings, some of which help explain existing data. I do however have a number of relatively modest requests for clarification of several points in order to make this work easier to understand and easier to place in the context of a number of other very recent papers on the same topic.

• The authors have made understanding some aspects of their model more difficult than necessary. In my opinion, they should move the torque model presentation from the SI to the main text and they should state much more clearly than is now the case that RNAP velocity is determined by torque difference.

• The authors should contrast the treatment of RNAP rotation here with that of other works in the literature. In particular, several papers have pointed out that the moment of inertia for RNAP and hence its rotational drag will increase as a result of RNA molecule growth and ribosome attraction, as the transcription process moves forward. The authors have not included this effect which may or may not be important; this issue should be discussed in the paper.

• The discussion of supercoiling diffusion is quite confusing. In fact, the authors use completely biased motion down the gradient which is hardly what one usually means by diffusion. I am not exactly sure how fixing this rate, which is more of a drag coefficient, can be compared to measurements of a diffusion rate; this point needs clarification.

• the statement on line 402, “we propose that intergenic supercoiling dynamics may mediate communication between neighboring genes” with no references given suggest that this is a new hypothesis. This should be corrected as it is not true. Also, as far as I can recall, the case of divergent promoters was studied by Kim et al and gave rise to lower transcription, in disagreement with the initiation-dominated result given here. Theoretical models do not have to agree with all aspects of the data in order to be interesting and useful, but disagreements should at least be discussed

• Finally, the authors are missing any mention of a new paper on this subject, “Collective polymerase dynamics …. “ by Sevier and Hornoz, bioarxiv (2021)

Reviewer #2: The work here presents a comprehensive quantitative model on supercoiling regulation in RNA polymerase (RNAP) transcription, incorporating the RNAP initiation, elongation, DNA supercoiling dynamics/diffusion, topoisomerase activities, and topological domain formation. The model is then implemented in biological context of multiple gene transcriptions with convergent, divergent, and co-directional multi-RNAP configurations, which lead to differently regulated collective behaviors of RNAPs for enhanced or inhibited productivities, due to the translocation-induced supercoiling and intergenic communication.

The model is essential in technically integrating RNAP transcription, supercoiling dynamics and topoisomerase activities altogether, providing a reasonable nice description of the overall process of supercoiling regulated transcription. Nevertheless, physical clarifications in the model construction appear to be lack of, in particular, on explicitly elucidating time scales of various sub-processes, which make the physical interpretations of current model not highly convincing. The model assumptions are not made sufficiently clearly in the beginning. In addition, as a quantitative model, various values of the parameter sets (e.g. different supercoiling diffusion rates, different topoisomerases) have not been tested, stability and/or robustness of the model or conclusions has not been well addressed. Therefore, the above major issues need to be resolved before this work being considered for publication.

Specific points for the authors’ consideration of improvements are listed below:

1) Page 7 line 65-67 on Kim et al [13]: how differently the RNAP initiation speeds vary under supercoiling comparing to without supercoiling? What is the measured promoter strength (e.g. how many RNAPs initiate per seconds)?

2) Figure 1 caption: several claims are made, such as “(A) a negatively supercoiled promoter is favored over a positively supercoiled promoter for RNAP binding and open complex formation” without references: pls either cite related literature or refer to specific text in current ms for further interpretations (and citations). Note that (B) for twin-domain supercoiling and (D,E) on Topo I and Gyrase all bear such issues (lack of references). Pls check similar lack of references for text elsewhere as well (e.g.,line 129-130 on RNAP footprint of 32 bp and Gyrase binding site 137 bp).

3) Modeling details: e.g. line 131-132 “to describe the spatial distribution of RNAP, topoisomerases, and supercoils at different positions along the DNA over time”: where are Topo I and Gyrase placed (upstream and downstream), and where are supercoiling (negative and positive) produced, and how fast they are generated (supercoiling generation) and persist (supercoiling relaxation), and how fast the supercoiling diffuses, with references, parameter values (not only provided in supplementary table) and justifications on the ranges.

4) Time scales and modeling assumptions in general: Line 136-138 “To describe the dynamics in the coupling of transcription and supercoiling, we focused on two questions: (1) how DNA supercoils are generated and dissipated by transcription and other processes, and (2) how transcription responds to DNA supercoiling”. The above questions are well raised, but poorly addressed/summarized in the model construction here. This is the right place to put up explicitly the time scales for supercoiling generation, dissipation, diffusion, and how fast the two typical types (?) of topoisomerases bind and unbind, which time scales are fast, which are slow, which are certain/ determined already, and which are uncertain and can be possibly tuned etc.. Meanwhile, it is the right place to list a summary of assumptions explicitly, though several assumptions spread around followed text.

5) Modeling Diffusion of Supercoiling and assumption: As the authors indicate (line 150-151) that twists and writhes diffuse differently along DNA. Even the writhes can also fast hop over a long distance (e.g. on ~1 �m/s from van Loenhout Science 2012). The authors assume that torsional stress diffuses relatively slowly, so only taking writhe diffusion because DNA exists mostly in the form of writhes under physiological conditions. This assumption surely leads to supercoiling density accumulation between adjacent RNAPs (FIG 2E) and two neighboring genes for codirectional case (FIG 4C). If assuming that the torsional stress diffuses as twists, which is much faster, between codirectional RNAPs and two genes, the positive and negative supercoiling may always cancel out fast. The authors only consider one limiting case for the torsional stress propagation. Right? If so, the authors should emphases this when drawing some conclusions (e.g., some conclusions related to FIG 2 and FIG 4.). Different assumptions lead to different conclusions.

6) Modeling diffusion as a biased random walk (line 166-167) to reduce computation cost: the authors should be cautious about this practice as free energy is introduced in the biased walk (as for molecular motor) such that the system operates in non-equilibrium; some of conclusions in regard to interaction of RNAPs with supercoiling may be impacted and should be re-examined.

7) Modeling topoisomerase binding/unbinding: it makes more sense to assume binding or on rate of proteins the same (i.e. diffusion limited), while dissociation or off rates vary as protein affinities with DNA differ. Besides, the binding/unbinding kinetic rates here should be tested in a range of values (compatible with experimental measurements or in a range reasonable in theory) to examine whether different values impact significantly on the RNAP-supercoiling dynamics or even lead to quite different working scenarios.

8) Modeling RNAP counter-rotation (line 214-215): Is the RNAP counter-rotation the only scenario that interprets Chong et al [8]’s measurements? What is the leaky diffusion of supercoil?

9) Results: Line 215-216, in Kim et al [13]: for promoter off case: if the last RNAP has no newly loaded RNAP (due to promoter off) to cancel its negative supercoiling upstream/behind, how could that reduce the forward elongation rate of RNAP? Are the modeling results always in agreement with the Kim et al [13]’s measurements and interpretations?

10) What is the elongation rate for a single RNAP in the same modeling context? How about the elongation rates vs the number of RNAPs loaded or in co-transcription case? The authors can show such plots for cooperative and antagonistic cases, respectively, or upon different initiation rates.

11) Results on cooperativity (line 295-297): why it is not the case that the new RNAP adapts to the slower rate of front RNAP (but font RNAP is pushed faster)? What parameters could directly impact on this cooperativity (or not)?

12) Results on “increased the Topo I activities” (line 304): are the binding or/and catalytic rates increased?

13) Figure 2D: what is the DNA supercoiling density for processive/paused RNAP: the supercoiling density on the DNA around RNAP? Why upstream can still have significant positive supercoiling density, while downstream can have significant negative supercoiling density (large initiation rate)?

14) Results on the mRNA production bursty on dynamical looping (line 362): does the number of RNAPs vary significantly? Does that contribute to the mRNA production busty or increased Fano factor? How exactly the dynamical looping was set in the simulation?

Reviewer #3: In this work, Gent et al. developed a comprehensive mathematical model to understand the roles of supercoiling in transcription regulation. They considered many regulations, including RNAP-supercoiling interactions, topoisomerase activities, stochastic topological domain formation, and supercoiling diffusion in all transcription stages. The model setup is very solid, with all the simplification, parameterization, and assumptions well justified by published data. Some of the predictions are very interesting. For example, the cooperative behavior of co-transcribing RNAP molecules could be a universal feature in E. coli gene transcription. Overall, the theoretical analysis in this work is very solid, and the predictions derived from the simulation are worthy of being tested experimentally.

Major.

1. For the Modeling supercoiling-sensitive transcription initiation section, is there any difference or revision of the equations compared to ref. 21? If not, please clearly state it. If yes, please clearly state the revision and difference.

2. The discussion of the supercoiling-dependent models in the design of synthetic gene circuits is very important and interesting. The simple two-gene system could yield complex behaviors due to the supercoiling dynamics. The authors could further expand this by discussing the hidden interaction between circuits and host cells (PMCID: PMC7870843, PMC7246135). For example, the resource competition for the ribosome and RNPA could lead to additional noise into the system but also constrain the noise level compared to the unlimited resource case (Advanced Genetics, 2022, 2100047). This is highly related to the additional transcription noise introduced in Fig 3C&F.

Minor.

3. Page 7, line 63, examples with citations are needed to clarify the emergent properties in transcription.

4. Citations are needed in line 99, page 9.

5. page 17, which figure is Fig. SX?

6. Refer fig. S12-13 in the main text as well. Is Fig. S11 missing?

7. Acknowledge the funding from National Science Foundation.

**Have the authors made all data and (if applicable) computational code underlying the findings in their manuscript fully available?**

Reviewer #1: Yes

Reviewer #2: None

Reviewer #3: Yes

PLOS authors have the option to publish the peer review history of their article (what does this mean?). If published, this will include your full peer review and any attached files.

Reviewer #1: No

Reviewer #2: No

Reviewer #3: No
---

## [Decision Letter · Decision Letter 1]

9 Jun 2022

Dear Ms Geng,

Thank you very much for submitting your manuscript "A spatially resolved stochastic model reveals the role of supercoiling in transcription regulation" for consideration at PLOS Computational Biology. As with all papers reviewed by the journal, your manuscript was reviewed by members of the editorial board and by several independent reviewers. The reviewers appreciated the attention to an important topic. Based on the reviews, we are likely to accept this manuscript for publication, providing that you modify the manuscript according to the review recommendations.

As you see, while two reviewers are satisfied with this revision, one reviewer still has several additional comments for you to consider by adding some discussions or clarifications. we hope that making these additional minor revisions can further improve the clarity of this paper.

Sincerely,

Chongzhi Zang

Guest Editor

PLOS Computational Biology

Ilya Ioshikhes

Deputy Editor

PLOS Computational Biology

[LINK]

As you see, while two reviewers are satisfied with this revision, one reviewer still has several additional comments for you to consider by adding some discussions or clarifications. we hope that making these additional minor revisions can further improve the clarity of this paper.

Reviewer's Responses to Questions

**Comments to the Authors:**

Reviewer #1: The authors have responded to all my previous requests for clarification and I am now in favor of acceptance

Reviewer #2: The revisions provide by the authors are substantial, and the overall work reads much more impressive than the first submission. The model is highly comprehensive and presented in well balance now. Most of my concerns have been addressed. There are some additional issues, however, that I recommend for the authors’ consideration to further clarify or comment:

1) The authors emphasize RNAP stall in Discussion (page 28), which has been suggested by Ma et al [ref10] from in vitro single molecule experiment. In Chong et al [ref 8], the initiation speed can be reduced approaching zero, while the elongation speed reduces but to about e.g. 1/3, or say, no stall appears for elongating RNAPs. Is there any scenario that explains such behaviors based on current model?

2) For the diffusion and twist: the authors modeled only the writhe diffusion (as the supercoiling diffusion), how about the twist? Does the twist diffuse differently? The supercoiling diffusion has been measured experimentally (van Loenhout et al ref [29]). In a recent modeling work on two-phase dynamics of supercoiling (Wan & Yu, Biophy J 2022), the diffusion of supercoiling is characterized via the boundary of the stretched phase (twist) and plectoneme (writhe), which may support effectively current treatment of diffusion. Meanwhile, the twist-writhe should together contribute to the toque, which current model does not seem to incorporate such level of physics details to calculate the torque.

3) In this model, it seems that the steady state behaviors are described, but how about the dynamic process of supercoiling generation (or relaxation or dissipation), e.g., how the supercoiling density builds up as the RNAP elongates? Does the supercoiling grow continuously, or discretely? In a previous model (Jing et al 2018 Phys Biol 15, 066007), it was assumed that the level of supercoiling builds up one-by-one (discretized) for every RNAP elongation such that multiple RNAP runs lead to multiple kinetic processes of supercoiling generation. Does the supercoiling dynamics demonstrated in current model suggest similar or different ideas? Besides, the authors do not show elongation rates for individual RNAPs, which if shown as a distribution would be nice to reflect the fluctuation behaviors of the convoy of RNAP.

In addition, in Fig 2D middle right, why the downstream supercoiling density is mainly negative for the processive RNAP (green ones)? Should the downstream supercoiling density to be dominantly positive for the transcribing RNAP?

Reviewer #3: The authors have addressed my comments.

**Have the authors made all data and (if applicable) computational code underlying the findings in their manuscript fully available?**

Reviewer #1: Yes

Reviewer #2: Yes

Reviewer #3: Yes

PLOS authors have the option to publish the peer review history of their article (what does this mean?). If published, this will include your full peer review and any attached files.

Reviewer #1: **Yes: **Herbert Levine

Reviewer #2: No

Reviewer #3: No

Figure Files:

Data Requirements:

Reproducibility:

References:

---

## [Decision Letter · Decision Letter 2]

12 Aug 2022

Dear Ms Geng,

We are pleased to inform you that your manuscript 'A spatially resolved stochastic model reveals the role of supercoiling in transcription regulation' has been provisionally accepted for publication in PLOS Computational Biology.

Best regards,

Chongzhi Zang

Guest Editor

PLOS Computational Biology

Ilya Ioshikhes

Deputy Editor

PLOS Computational Biology

Reviewer's Responses to Questions

**Comments to the Authors:**

Reviewer #2: The authors have well addressed my concerns. It's a highly impressive work, and I gladly recommend it for publication!

**Have the authors made all data and (if applicable) computational code underlying the findings in their manuscript fully available?**

Reviewer #2: None

PLOS authors have the option to publish the peer review history of their article (what does this mean?). If published, this will include your full peer review and any attached files.

Reviewer #2: No

---

## [Editor Report · Acceptance letter]

2 Sep 2022

PCOMPBIOL-D-21-02311R2 

A spatially resolved stochastic model reveals the role of supercoiling in transcription regulation

Dear Dr Geng,

I am pleased to inform you that your manuscript has been formally accepted for publication in PLOS Computational Biology. Your manuscript is now with our production department and you will be notified of the publication date in due course.

With kind regards,

Olena Szabo
